# Subsequent Systemic Therapy following Platinum and Immune Checkpoint Inhibitors in Metastatic Urothelial Carcinoma

**DOI:** 10.3390/biomedicines10082005

**Published:** 2022-08-18

**Authors:** Joohyun Hong, Hyun Hwan Sung, Byong Chang Jeong, Se Hoon Park

**Affiliations:** 1Division of Hematology and Oncology, Department of Medicine, Samsung Medical Center, Sungkyunkwan University School of Medicine, Seoul 06351, Korea; 2Department of Urology, Samsung Medical Center, Sungkyunkwan University School of Medicine, Seoul 06351, Korea

**Keywords:** urothelial carcinoma, chemotherapy, salvage therapy, clinical trials

## Abstract

Treatment of metastatic urothelial carcinoma (mUC) after failure with platinum-based chemotherapy and immune checkpoint inhibitors (ICIs) remains controversial. To explore the role of subsequent systemic therapy, medical records from 436 patients who were consecutively treated with chemotherapy for mUC between May 2017 and April 2021 were collected from a single-center cancer registry. The primary endpoint was overall survival (OS), and progression-free survival (PFS) and response rate (RR) were also assessed. Among the 318 patients who failed both platinum and ICIs, subsequent therapy was delivered to 166 (52%) patients: taxanes (*n* = 56), platinum rechallenge (*n* = 46), pemetrexed (*n* = 39), and clinical trials (*n* = 25). Objective responses to third-line therapy were noted in 50 patients (RR, 30%; 95% CI, 23–37%). The patients who were enrolled in clinical trials and treated with platinum rechallenge were significantly more likely to respond than those treated with taxanes or pemetrexed. The median PFS and OS were 3.5 months (95% CI, 2.9–4.2 months) and 9.5 months (95% CI, 8.1–11.0 months), respectively. Similar to RR, PFS and OS were longer for the patients who were enrolled in clinical trials. Based on multivariate analyses, good performance status and enrollment in clinical trials are associated with benefits from subsequent therapy for pretreated mUC.

## 1. Introduction

Over the past two decades, platinum-based combination chemotherapy remains the standard of care for patients with previously untreated metastatic urothelial carcinoma (mUC). Clinical trials testing cisplatin-containing first-line chemotherapy regimens demonstrated a median overall survival (OS) of 14 to 15 months [1]. For patients who are medically “unfit” or ineligible to receive cisplatin due to poor performance status, impaired renal function, or co-morbidities [2], a median OS of 8 to 9 months can be achieved with carboplatin-containing chemotherapy [3]. If a patient experiences disease progression during or following first-line platinum-based chemotherapy, second-line therapy with programmed death-1 (PD-1) and programmed death-ligand 1 (PD-L1) immune checkpoint inhibitors (ICIs), such as pembrolizumab [4], is recommended based on phase III clinical trials. Another anti-PD-L1 ICI, atezolizumab, was initially included in the guidelines as a second-line treatment. However, atezolizumab failed to confer an OS benefit over chemotherapy in a randomized phase III trial [5], leading to the withdrawal of regulatory approval for this indication. More recently, another ICI, avelumab, has had regulatory approval for first-line maintenance therapy following disease control (i.e., clinical responses or stable disease) with four to six cycles of platinum-based chemotherapy in patients with mUC [6]. It is of note that the OS benefit achieved with avelumab maintenance following chemotherapy was encouraging in both the total population (hazard ratio (HR), 0.69; 95% confidence interval (CI), 0.56 to 0.86) as well as in those with a PD-L1 overexpression (HR, 0.56, 95% CI, 0.40 to 0.79).

Despite these advancements with the introduction of ICIs, a significant percentage of patients would develop resistance to ICIs after months, or even years, of disease control. However, due to the lack of evidence associated with the benefit of third- or subsequent lines of therapy, and the potential for toxicity from such treatment, the proportion of the patients offered further therapy varies from 30% to 50% [7,8]. In general, chemotherapy in mUC should be administered to prolong survival and improve the quality of life of the patients, a factor that is even more important in salvage settings. As randomized, controlled clinical trials are sparse in this setting, we performed the present retrospective analysis to develop improved therapeutic strategies for patients with mUC, enhance patient counseling, and generate hypotheses for future studies.

## 2. Materials and Methods

With the help of the Samsung Medical Center (SMC, Seoul, Korea) cancer registry, individual patient-level structured data for adult (>20 years of age) mUC patients treated with first-line chemotherapy were collected and reviewed. The inclusion criteria for the present retrospective study included: (1) histologically proven diagnosis of UC arising from the bladder and/or upper urinary tract, (2) the presence of measurable metastatic disease, (3) treated with third-line therapy following failure to first-line platinum-based chemotherapy and second-line ICIs, and (4) the availability of clinical data at the beginning of therapy and follow-up. Written informed consent was given by all patients prior to receiving treatment, according to institutional guidelines. This retrospective study was reviewed and approved by the SMC institutional review board (SMC IRB No. 2018-02-016).

The decision for administering subsequent systemic therapy following platinum and ICIs was, in all cases, at the discretion of the medical oncologists. The third-line therapy regimen to be used was determined by the treating physician but, in some patients, in the context of clinical trials. All tumor measurements were assessed every 2 or 3 months after starting therapy, by using abdominopelvic computed tomography (CT) scan and other tests that were initially used to stage the tumor. Tumor response was evaluated and reviewed by an investigator (J.H.) at the time of analysis, according to the Response Evaluation Criteria for Solid Tumors (RECIST) v1.1. The primary endpoint of this study was OS. The secondary endpoints included progression-free survival (PFS) and response rate (RR). The starting point of OS and PFS was the first day of third-line therapy. The date of disease progression or death from causes other than mUC was used in calculating PFS. Time of death, whatever the cause, was used to calculate OS. PFS and OS were estimated according to the Kaplan–Meier method, and the median values and 95% confidence intervals (CIs) were calculated. A Cox proportional hazard regression model was used to identify independent clinical and treatment factors associated with prolonged OS. Factors for regression analysis included age (below vs. ≥median), gender, primary sites (upper tract vs. bladder), an Eastern Cooperative Oncology Group (ECOG) performance status (0–1 vs. ≥2), the best response to first-line chemotherapy, the number of involved sites (one vs. ≥2), metastases (lymph node only vs. bone, visceral disease), baseline chemistry profiles (serum albumin, calcium and hemoglobin levels), and therapy regimens. Laboratory parameters were initially recorded as continuous variables and later dichotomized according to the mean value of each variable (below vs. ≥mean). All *p* values were two-sided, with *p* < 0.05 indicating statistical significance.

## 3. Results

Medical records from 436 eligible patients who were consecutively treated with first-line platinum-based combination chemotherapy for mUC at the medical oncology department of SMC between May 2017 and April 2021 were collected for the present retrospective study (Figure 1). Overall, 73% (*n* = 318) of the patients were treated with cisplatin-based chemotherapy, while others received carboplatin. With the median follow-up of 40 months, the median OS for all the 436 patients was calculated as 17.8 months (95% CI, 15.7 to 19.9 months). After the completion of first-line chemotherapy, 318 (73%) patients received second-line ICIs involving atezolizumab (*n* = 255), pembrolizumab (*n* = 35), nivolumab (*n* = 21), and durvalumab (*n* = 7). The estimated median OS was significantly longer for the patients who received second-line ICIs (20.8 months; 95% CI, 17.7 to 23.9 months) than those who did not (9.0 months; 95% CI, 4.3 to 11.7 months).

Among the 318 patients that progressed after treatment with platinum and ICIs, subsequent third-line therapy was delivered to 166 (52%) patients. Most (79%) patients were male, and the median age was 67 years (range, 33–86 years). Overall, 77% of the patients had visceral (lung and/or liver) metastases, and 18% had bone involvement. Furthermore, 17% of the patients with an ECOG performance status of 2 received third-line therapy. The baseline patient characteristics for these 166 patients are given in Table 1. Although we did not evaluate individual-level safety data, four possible treatment-related deaths were identified. One sudden death occurred during the first cycle of pemetrexed, without any clinical evidence of disease progression having been demonstrated. Three other deaths were recorded in patients while receiving taxanes, which were attributed to neutropenic sepsis.

Of the 166 patients who were treated with third-line therapy, 56 patients (one-third of the total) were treated with taxane monotherapy, 46 received platinum rechallenge, 39 received pemetrexed, and others received novel therapeutics in the context of clinical trials. The most commonly delivered clinical trial drug was enfortumab vedotin (*n* = 13), followed by erdafitinib (*n* = 8) and others (*n* = 4). Of the 166 patients included in this analysis, 5 could not be evaluated for responses because of early discontinuation of therapy. Objective responses to third-line therapy were noted in 50 patients (RR, 30%; 95% CI, 23 to 37%), out of which 9 patients had complete responses. The patients who were enrolled in clinical trials and treated with platinum rechallenge (RR, 52% and 37%, respectively) were significantly more likely to respond than those treated with taxanes (21%) or pemetrexed (21%). Notably, among the 25 clinical trial participants, complete and partial responses were noted in 6 and 7 patients, respectively (Table 2). Another factor associated with higher RR was a prior response to first-line chemotherapy (26% vs. 9%; *p* = 0.016). RR was not influenced by age, gender, primary sites, baseline laboratory parameters, performance status, or metastatic sites.

Of the 166 patients included in the present analysis, 117 (71%) died as of the data cutoff (Jan 2022). The estimated median PFS was 3.5 months (95% CI, 2.9 to 4.2 months), and the median OS was 9.5 months (95% CI, 8.1 to 11.0 months; Figure 2). Similar to RR, PFS was longer for patients who were enrolled in clinical trials (Table 2). In the univariate model, the estimated OS was significantly shorter for patients with low baseline hemoglobin, poor performance status, no response to first-line chemotherapy, and those treated with pemetrexed. We tested whether the OS was modified by interactions between the effect of the significant factors and the third-line regimens; the first-level interaction term between these factors was entered into separate multivariate Cox regression models. The interaction between performance status and hemoglobin level was the only significant interaction (correlation coefficient, −0.54; *p* = 0.04). In a multivariate one-stage random effects Cox regression model, performance status (*p* < 0.01) was a significant factor for longer OS in addition to third-line clinical trials (*p* = 0.03). Patients with a performance status of 0 to 1, compared with 2 (HR, 0.23; 95% CI, 0.15 to 0.35), had an improved OS. Likewise, the patients who were enrolled in clinical trials as third-line therapy were more likely to have longer OS than those treated with taxanes, platinum, or pemetrexed (median, 18.0 vs. 8.9 months; HR, 0.51, 95% CI, 0.27 to 0.96). Further fourth or more lines of therapy was delivered to 63 (38%) patients after third-line failure. OS was similar between the patients who received the fourth line of therapy and those not treated further (median, 9.8 vs. 9.2 months; *p* = 0.17).

## 4. Discussion

The present retrospective study of 166 mUC patients who were treated with subsequent anticancer systemic therapy after failure to both platinum and ICIs showed a strong association between the baseline performance status and prolonged OS. The study may provide important guidance for the treatment of patients with pretreated mUC. In addition, our study emphasized that clinical trials are an essential part of cancer treatment, especially for patients who have no proven options. Overall, 15% of our patients participated in clinical trials involving enfortumab vedotin, erdafitinib, or others; these drugs are now considered a standard of care for mUC in this setting. Novel therapeutics other than ICIs have recently emerged for patients who fail after platinum and ICIs, including erdafitinib and enfortumab vedotin. Erdafitinib is an orally available fibroblast growth factor receptor (FGFR) inhibitor and was granted FDA approval in 2019 for patients with mUC who progress during or after platinum-based chemotherapy [9]. In an updated publication of the BLC2001 phase II study [10], second-line erdafitinib therapy for patients with mUC and prespecified FGFR alterations showed a RR of 40% (95% CI, 30 to 49%) and PFS of 5.5 months (95% CI, 4.3 to 6.0 months). Erdafitinib is being tested in a phase III randomized, controlled trial (ClinicalTrials.gov, NCT03390504) in patients with mUC, compared with ICIs or chemotherapy. Enfortumab vedotin is an antibody-drug conjugate directed to nectin-4 [11] and was reported to yield a RR of 52% (95% CI, 41 to 62%) and a median PFS of 5.8 months (95% CI, 5.03 to 8.28 months). In a subsequent phase III randomized trial comparing enfortumab vedotin with chemotherapy in a salvage setting, significantly longer OS (median, 12.88 vs. 8.97 months) was reported. The US Food and Drug Administration (FDA) has approved erdafitinib and enfortumab vedotin for post-chemotherapy mUC settings.

Unfortunately, despite these novel therapeutics developed for the treatment of mUC, eligible FGFR3 alterations have been reported in less than 20% [9], and a significant proportion of patients are ineligible for these drugs or subsequently progress. Furthermore, at present, neither enfortumab vedotin nor erdafitinib has been approved in Korea, and no standard third-line or salvage therapy is available for patients with mUC and failure to platinum and ICIs. In this setting, many patients experience adverse events and symptoms, leading to rapid clinical deterioration and chemotherapy ineligibility. Nevertheless, it is a common practice to offer third-line therapy only if the patients continue to have a good performance status and are still medically fit enough to receive further therapy, probably because patients and physicians have difficulty in accepting only supportive care without the possibility of systemic anticancer effects. Although it is recognized that there is a declining probability of response to subsequent therapy, small phase II studies involving salvage chemotherapy with taxanes or pemetrexed have resulted in an objective RR of approximately 10% and a median OS of 7 to 9 months [12,13,14,15,16]. Multi-drug combination chemotherapy may achieve a longer OS than monotherapy [16], along with a significantly higher risk of severe hematologic toxicities.

Clearly, third-line therapy may not be beneficial for all patients and there is potential for toxicity from the treatment. The administration of an active and tolerable chemotherapy regimen in a well-selected patient population would lead to an improvement in clinical outcomes. Therefore, it is necessary to better define the subsets of patients who may benefit from further therapy, and the identification of factors allowing the selection of these patients is an important challenge. We found that performance status emerged as a significant predictor for OS. Poor performance status is a common finding among mUC patients who experienced disease progression after platinum-based chemotherapy and second-line therapy involving ICIs. When interpreting the results, it is of note that the present analyses represent only a small sample of patients, and we cannot completely exclude the possibility that poor performance status may be reflective of an extensive disease burden and/or other predictors for a poor prognosis. It is indicated that most clinical trials exclude such patients, who may rarely benefit from salvage therapy. Besides clinical factors, appropriate patient selection based on the molecular or genomic landscape is one of the most extensively studied areas in clinical research. UC is a genomically heterogeneous disease [17], with significantly mutated genes and high frequencies of occurrence of several essential pathways regulating chromatin state, cell-cycle regulation, and receptor kinase signaling such as the MAPK, PI3K/AKT, FGFR/RAS, and TP53/RB1/MDM2, RAP1 pathways closely related to tumor progression and tumor evolution. It still is at an early stage to understand the complexity of the clinical sequencing data [18,19], and molecular biomarker studies failed to comprehensively identify patients who could benefit from therapy [20]. One of the examples of novel therapeutics directed to specific molecular targets included erdafitinib for mUC patients with prespecified FGFR alterations [9]. Similarly, we have conducted a phase II study of a farnesyltransferase inhibitor tipifarnib in mUC patients whose tumors harbor missense HRAS mutations [21]. Tipifarnib was generally well-tolerated and resulted in a RR of 33% in heavily pretreated mUC patients with HRAS mutations. Unfortunately, mutations in FGFR or HRAS are found in only a few cases with mUC [17,21], and an adaptive, biomarker-directed platform study of ICIs in combination with different targeted therapies in mUC failed to meet its efficacy criteria for further development [22]. Screening for genetic aberrations by using whole-exome sequencing or multiplexed targeted DNA sequencing is now broadly conducted across many cancer types including mUC, and it may increase the likelihood of finding patients to benefit from such molecularly targeted therapies.

If a patient with pretreated mUC would receive third- or salvage therapy, a decision on chemotherapy regimens for an individual patient may be a common clinical situation. For subsequent therapy after platinum-based chemotherapy and ICIs, the preferred therapeutic options include enfortumab vedotin or erdafitinib. It is interesting that some of our patients had received rechallenge platinum-based chemotherapy. The RR and PFS observed with platinum rechallenge are promising, compare favorably to those achieved with taxanes or pemetrexed [12,13,14], and are consistent with previously reported retrospective studies of platinum rechallenge in mUC [23,24,25]. Platinum agents, either cisplatin or carboplatin for cisplatin-ineligible patients, have still been established as important drugs in the palliative setting of mUC. At present, nobody knows the best drug or regimen for subsequent systemic therapy following failure to platinum and ICIs because of the limitation of comparison between different studies. While seeking the answers to these questions, we should keep in mind that the role of subsequent therapy in this setting remains strictly palliative, in terms of its ability to substantially improve OS and maintain patients’ quality of life. Despite recent advances, the prognosis of patients with mUC remains poor. Since not all patients with mUC are eligible to receive platinum-based chemotherapy as third-line therapy, and enfortumab vedotin or erdafitinib has not yet been approved in Korea, it is generally recommended to treat these patients within clinical trials.

## Figures and Tables

**Figure 1 biomedicines-10-02005-f001:**
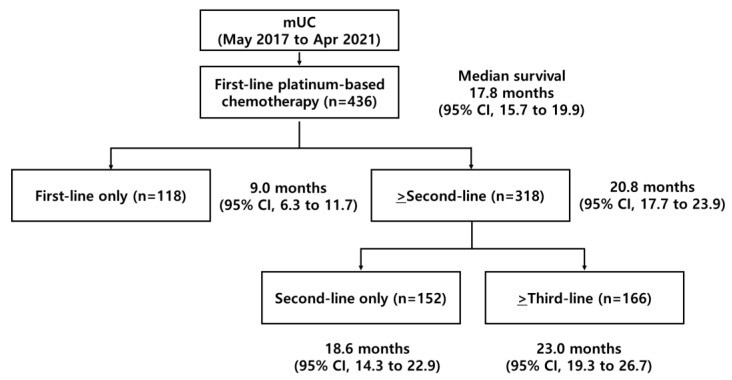
Diagram and median overall survivals for patients with metastatic urothelial carcinoma (mUC) treated between May 2017 and April 2021.

**Figure 2 biomedicines-10-02005-f002:**
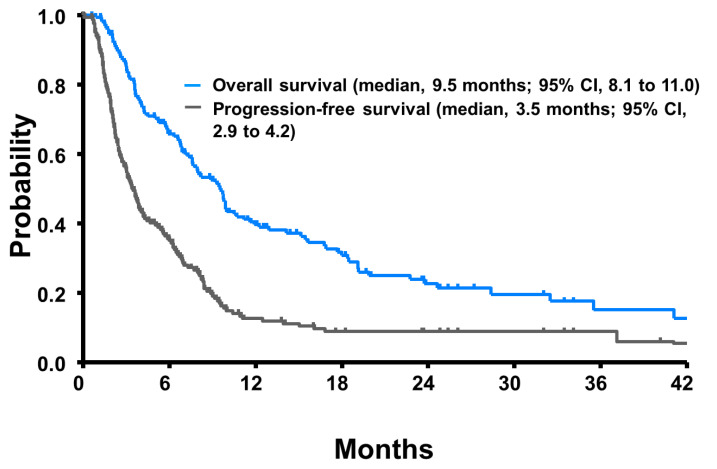
Progression-free survival (gray line) and overall survival (blue line) for 166 patients who were treated with third-line therapy.

**Table 1 biomedicines-10-02005-t001:** Patient characteristics at the time of third-line therapy.

	No. of Patients (*n* = 166)
Age, years	
Median	67
Range	33–86
Gender	
Male	131 (79%)
Female	35 (21%)
Primary site	
Upper tract	79 (48%)
Bladder	87 (52%)
First-line platinum	
Cisplatin	131 (79%)
Carboplatin	35 (21%)
Second-line checkpoint inhibitor	
Atezolizumab	148 (89%)
Pembrolizumab	16 (10%)
Nivolumab	2 (1%)
Response to first-line chemotherapy	
Responder	87 (52%)
Stable disease	30 (18%)
Progressive disease or unknown	49 (30%)
ECOG performance status	
0 to 1	138 (83%)
2 or more	28 (17%)
Metastatic site(s)	
Lymph node only	29 (18%)
Visceral (lung and/or liver) metastases	127 (77%)
Bone metastasis	30 (18%)
Baseline laboratory values, mean	
Hemoglobin, g/L	9.9
Albumin, g/dL	3.4
Calcium, mg/dL	8.7

ECOG denotes Eastern Cooperative Oncology Group.

**Table 2 biomedicines-10-02005-t002:** Outcomes of third-line therapy according to regimens.

	Taxanes (*n* = 56)	Platinum (*n* = 46)	Pemetrexed (*n* = 39)	Clinical Trials (*n* = 25)
Clinical responses				
Complete	1	1	0	6
Partial	11	16	8	7
Stable disease	5	8	4	5
Progression	39	21	27	7
Progression-free survival				
Median, mo	2.4	5.8	2.9	8.5
95% CI	2.1 to 2.8	2.8 to 8.8	1.2 to 4.6	4.3 to 12.6
Overall survival				
Median, mo	9.6	10.0	6.3	18.0
95% CI	8.4 to 10.7	8.4 to 11.5	3.9 to 8.8	8.8 to 27.2

## Data Availability

Not applicable.

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
