# Peer review of "Subsequent Systemic Therapy following Platinum and Immune Checkpoint Inhibitors in Metastatic Urothelial Carcinoma"

_biomedicines, 2022, doi:10.3390/biomedicines10082005_

Round 1

Reviewer 1 Report

This is a well-written and well-planned retrospective study of patients with mUC who progressed on at least 2 lines of therapy. This is very important real-world data from Korea that should be published. The authors point out that patients who received a 3rd line of therapy that was part of a clinical trial did better than those who received other types of therapy, and those with good ECOG should get at least a 3rd line. The figures and tables complement the text well. Overall, the authors should be commended on their work.

1. I would recommend a flow diagram (if data is available) as part of Figure 1 (or a separate figure) determining how the total number of patients in the cohort became whittled down to N=436.

2. There are several minor grammar/English flaws so the manuscript text needs a careful re-read for edits, some of which include:

Line 15 remove "to"

Line 36 is should be replaced with are

Line 100 remove "to"

Line 110 should say one-third without the s

Line 111 remove s from taxanes

Line 157 add s to emphasize

Line 162-164 sentence is too long and grammatically incorrect

Line 225 remove s from patients

Line 230 remove s from compares

Line 231 needs "are" before consistent

Author Response

This is a well-written and well-planned retrospective study of patients with mUC who progressed on at least 2 lines of therapy. This is very important real-world data from Korea that should be published. The authors point out that patients who received a 3rd line of therapy that was part of a clinical trial did better than those who received other types of therapy, and those with good ECOG should get at least a 3rd line. The figures and tables complement the text well. Overall, the authors should be commended on their work.

  1. I would recommend a flow diagram (if data is available) as part of Figure 1 (or a separate figure) determining how the total number of patients in the cohort became whittled down to N=436.

-> Thanks for the favorable review. In fact, we collected data of consecutively treated patients with mUC from our cancer registry, and identified 436 patients. Who met the inclusion criteria. Of course we had more patients including those who did not receive platinum-based combination chemotherapy but, unfortunately, the data are not available.

  1. There are several minor grammar/English flaws so the manuscript text needs a careful re-read for edits, some of which include:

Line 15 remove "to"

Line 36 is should be replaced with are

Line 100 remove "to"

Line 110 should say one-third without the s

Line 111 remove s from taxanes

Line 157 add s to emphasize

Line 162-164 sentence is too long and grammatically incorrect

Line 225 remove s from patients

Line 230 remove s from compares

Line 231 needs "are" before consistent

-> Thanks for invaluable suggestions. The paper was edited according to your comments.

Reviewer 2 Report

Introduction section, line 36, "(...), such as atezolizumab or pembrolizumab (4,5), is recommended based on phase III clinical trials"

Atezolizumab is not currently recommended as second line tretament in mUC.

Accordingly with NCCN bladder cancer guidelines:

"In March 2021, the makers of atezolizumab voluntarily withdrew their indication for patients with locally advanced or metastatic urothelial carcinoma that was previously treated with a platinum-based chemotherapy.272 This withdrawal was based on the IMvigor211 trial failing to meet its primary endpoint of improved OS. Therefore, the NCCN Panel does not recommend atezolizumab as a second-line option following platinum-based therapy, although it is still recommended in its first-line indication."

Suggestion: It would be better to remove atezolizumb from line 36. 

Results section, line 100, "Among 318 patients who failed to both platinum and ICI, subsequent (...)"

Suggestion: Among 318 patients that progressed after treatment with platinum and ICI, subsequent (...)

line 110 "(...) with third-line therapy, one-thirds of patients (...)

Suggestion: (...) with third-line therapy, one-third of patients(...)

line 118 "(...)with platinum rechallenge (52 % and 37 %, respectively) were significantly(...)"

Suggestion (...)with platinum rechallenge (RR 52 % and 37 %, respectively) were significantly(...)"

Author Response

Introduction section, line 36, "(...), such as atezolizumab or pembrolizumab (4,5), is recommended based on phase III clinical trials"

Atezolizumab is not currently recommended as second line tretament in mUC.

Accordingly with NCCN bladder cancer guidelines:

"In March 2021, the makers of atezolizumab voluntarily withdrew their indication for patients with locally advanced or metastatic urothelial carcinoma that was previously treated with a platinum-based chemotherapy.272 This withdrawal was based on the IMvigor211 trial failing to meet its primary endpoint of improved OS. Therefore, the NCCN Panel does not recommend atezolizumab as a second-line option following platinum-based therapy, although it is still recommended in its first-line indication."

Suggestion: It would be better to remove atezolizumb from line 36.

-> We do agree with your comment. Atezolizumab was removed, and we edited the sentence as “Another anti-PD-L1 ICI, atezolizumab was initially included in the guidelines as a second-line treatment. However, atezolizumab failed to confer an OS benefit over chemo-therapy in a randomized phase III trial [5], leading to withdrawal of regulatory approval for this indication.” Thanks.

Results section, line 100, "Among 318 patients who failed to both platinum and ICI, subsequent (...)"

Suggestion: Among 318 patients that progressed after treatment with platinum and ICI, subsequent (...)

-> The sentence was revised accordingly.

line 110 "(...) with third-line therapy, one-thirds of patients (...)

Suggestion: (...) with third-line therapy, one-third of patients(...)

line 118 "(...)with platinum rechallenge (52 % and 37 %, respectively) were significantly(...)"

Suggestion (...)with platinum rechallenge (RR 52 % and 37 %, respectively) were significantly(...)"

  • Thanks.